# Effects of Acute Aquatic High-Intensity Intermittent Exercise on Blood Pressure and Arterial Stiffness in Postmenopausal Women with Different ACE Genotypes

**DOI:** 10.3390/ijerph19158985

**Published:** 2022-07-23

**Authors:** Wen-Sheng Zhou, Ai-Lun Yang, Chiao-Nan Chen, Nai-Wen Kan, Joanna Ting-Hui Kuo, Lee-Hwa Chen, Kuei-Yu Chien

**Affiliations:** 1Department of Physical Education, Nanjing Xiao-Zhuang University, Nanjing 211171, China; zhouwensheng@njxzc.edu.cn; 2Graduate Institute of Sports Science, National Taiwan Sport University, Taoyuan 333325, Taiwan; 1050213@ntsu.edu.tw; 3Institute of Sports Sciences, University of Taipei, Taipei 11153, Taiwan; alyang@utaipei.edu.tw; 4Department of Physical Therapy and Assistive Technology, School of Biomedical Science and Engineering, National Yang Ming Chiao Tung University, Taipei 112032, Taiwan; cnchen@nycu.edu.tw; 5Center for General Education, Taipei Medical University, Taipei 11031, Taiwan; kevinkan@tmu.edu.tw; 6Department of Athletic Training and Health, National Taiwan Sport University, Taoyuan 333325, Taiwan; lhchen@ntsu.edu.tw

**Keywords:** middle-aged and elderly women, gene polymorphism, angiotensin-converting enzyme, artery elasticity, high-intensity intermittent exercise

## Abstract

The present study investigated the effects of acute aquatic high-intensity intermittent jumping (HIIJ) on blood pressure (BP) and arterial stiffness in postmenopausal women with different angiotensin-converting enzyme genotypes (ACE). We recruited 12 postmenopausal women carrying the ACE deletion/deletion (DD) genotype and 61 carrying the insertion/insertion or insertion/deletion (II/ID) genotype. The participants performed 12 trials of 30 s, 75% heart rate reserve (HRR) jumping, and 60 s, 50% HRR recovery, and 3 trials of 40 s upper limb resistance exercises were performed as fast as possible. The heart rate (HR) and BP were measured before exercise, immediately, 10 min, and 45 min after exercise. The brachial-ankle pulse wave velocity (baPWV) was measured before and after exercise. The systolic blood pressure (SBP) of the DD genotype increased more significantly than those with the II/ID genotype post-exercise (30.8 ± 4.48 vs. 20.4 ± 2.00 mmHg, *p* = 0.038). The left and right sides of baPWV increased significantly after exercise (1444.8 ± 29.54 vs. 1473.4 ± 32.36 cm/s, *p* = 0.020; 1442.1 ± 30.34 vs. 1472.0 ± 33.09, *p* = 0.011), and there was no significant difference between the two groups. The HIIJ increased baPWV. The postmenopausal women with the DD genotype have a higher SBP increased post-exercise than those with II/ID genotype. These findings suggest that the aquatic exercise program has better effects in decreasing blood pressure in postmenopausal women with the II/ID genotype. Those with the DD genotype should pay attention to the risk of increasing blood pressure after aquatic HIIJ exercise.

## 1. Introduction

Hypertension (HTN) and arterial stiffness (AS) are common cardiovascular and cerebrovascular diseases that are the main risk factors for mortality [1,2]. Increased blood pressure and increased AS in postmenopausal women are related to decreased secretion of estrogen [3,4]. Studies have demonstrated that with every 10 mmHg decrease in systolic blood pressure (SBP), the risk of cardiovascular disease is reduced by 20%, and the overall mortality rate is reduced by 13% [5,6]. Arterial stiffness can cause heart failure, stroke, and kidney diseases [7,8], and it is an early indicator of cardiovascular disease prediction. The brachial-ankle pulse wave velocity (baPWV) is an independent predictor of cardiovascular diseases’ incidence rate and mortality [9].

Previous studies have demonstrated that a single exercise on land causes post-exercise hypotension (PEH). The PEH is the main mechanism for improving BP [10,11]. The participation of postmenopausal women in exercises was restricted due to joint diseases, insufficient muscle strength, and fear of falling [12]. Aquatic exercise can circumvent these restrictions, with buoyancy reducing 50–90% of the impact on joints [13,14], helping overcome the fear of falling and ensuring safety [15]. However, the acute effect of exercise on BP remains controversial. Cunha et al. (2016) have suggested that older women excited a 45-min moderate-to-high intensity aquatic exercise, then resting for 21 h, showed an average reduction in SBP and diastolic blood pressure (DBP) of approximately five and one mmHg, respectively [16]. Lai et al. (2015) have observed that SBP decreased by 7.3%, and DBP had no change 9 h after a 25-min aquatic walking exercise with 70% peak oxygen uptake (VO_2peak_) [17]. Rodriguez et al. (2011) studied a 45-min aquatic brisk walking with 40% VO_2peak_ and found that the average SBP during 30-60 min after exercise and the average DBP during 45–60 min after exercise of untrained women decreased by 9.3% and 8.7%, respectively [18]. For trained women, the average SBP during 15–60 min after exercise was significantly reduced by 10%, and there was no change in DBP. However, it is also widely known that the inconsistent PEH results were not only related to exercise intensity, duration, and health status of all individuals [19,20,21] but also might partly be explained by genetic factors [22,23]. Previous studies have concluded that genetic factors explain up to 17% of training-induced SBP reduction [22,23]. The angiotensin-converting enzyme (ACE) genotype is an important genetic factor that is responsible for BP regulation in the renin-angiotensin-aldosterone system [24,25].

ACE can convert angiotensin I (Ang I) into powerful angiotensin II (Ang II), increasing BP [26,27]. The polymorphism of ACE includes insertion (I), and deletion (D), and accordingly, ACE is classified into II genotype, ID genotype and DD genotype. Previous studies have indicated that the percentage of ACE DD genotype is 7.0–14.7% in Asia, and that of II/ID genotype is 85.3–93% [28,29,30,31,32]. Individuals with the DD genotype have higher ACE circulating levels than those with the II genotype [33], leading to elevated BP and HTN risk [34,35,36]. At present, few studies have explored the effects of ACE on PEH. Previous studies have shown that there was different blood pressure altered in the people with different ACE genes after exercise [34,37,38]. Freire et al. (2015) found that the SBP of people with the DD genotype at a resting state was significantly higher than those with the II or ID genotype [37]. It has also been found that resistance exercises significantly increased the SBP and DBP of people carrying the DD genotype [37]. Goessler et al. (2015) confirmed that the SBP and DBP of people with the II and ID genotypes were significantly decreased 1 h after a single walk with 60–75% HRR while those of people carrying DD genotype did not change [34]. Santana et al. (2011) confirmed that after performing a maximal incremental exercise test and exercise with a 90% anaerobic threshold, elderly women with II or ID genotype had a more significant decrease in SBP within 1 h than those with DD genotype and there was no difference in DBP [38]. The effects of exercises on AS are controversial. Some studies have shown that arterial PWV increased by 4.3–20.8% immediately after moderate-to-high-intensity exercises [39,40,41,42], while some studies have confirmed that arterial PWV decreased by 0.08–9.4% immediately after moderate-to-high-intensity exercises [43,44]. There was also a study showing that moderate-intensity exercises did affect arterial PWV [43]. In the aquatic environment study, Sosner et al. (2016) demonstrated that 24-h aortic PWV was significantly decreased after aquatic high-intensity intermittent exercise (HIIE) with immersed cycling (up-to-the-chest) while the effects were not found after HIIE on dryland [45]. Viana et al. (2019) also confirmed that the carotid-femoral PWV was no change at the post 45 min following dryland HIIE in Type 2 diabetes [46]. 

To our knowledge, there is no research on the effects of the ACE genotype and acute aquatic high-intensity intermittent jumping (HIIJ) on AS. We hypothesized that, compared to the II/ID genotype, postmenopausal women with the DD genotype would obtain more beneficial effects for improving BP and AS after exercise. The purpose of the present study aimed to investigate the effects of an acute aquatic HIIJ on BP and AS in postmenopausal women with different ACE genotypes.

## 2. Materials and Methods

### 2.1. Participants

One hundred and eighty-one postmenopausal women were recruited to participate in the experiencing lesson on aquatic exercise. Participants included in the present study need to have been through menopause for more than one year. The exclusion criteria were as follows: participants have recently taken ACE inhibitor drugs, have been hospitalized within three months, have uncontrollable HTN, metabolic diseases, and neuromuscular diseases, respiratory diseases, skin diseases, movement problems, fear of water, or cognitive impairment. We do not know who had which type of ACE polymorphisms, therefore, we assessed ACE polymorphism after recruitment. Among the recruited participants, 27 were excluded, and a total of 154 attended the experiencing lesson and did genotype testing. A total of 81 participants were excluded because of unsatisfying exercise performance, HTN and joint discomfort during the experiencing lesson. Seventy-three participants (62.3 ± 6.05 years, BMI: 23.3 ± 2.37 kg/m^2^) were included in the formal exercise lesson with 12 carrying ACE DD genotype and 61 carrying II/ID genotype. The participants were told to maintain their daily routines and not engage in strenuous exercise three days before the experiment. Participants with HTN took medicines before exercise. Participants signed an institutionally approved informed consent form after fully understanding the content of the study. This research was approved by the Institutional Review Board of Fu Jen Catholic University (Approval number: C106070).

### 2.2. Study Procedures

The participants arrived at the experimental site and sat for 5 min on land then measured BP, HR and RPE. The participants filled in an individual health questionnaire about exercise behavior, hypertension, diseases of cardiovascular, respiratory, digestive, endocrine, bone, joints, ear, eye, and cancer. They performed the PAR-Q questionnaire and body compositions. The oral mucosa of participants was collected to analyze the ACE genotype. The participants were invited into a quiet room for baPWV measurement. The HR and BP after 5-min stand resting on land (SRL, near the swimming pool) and after 5-min stand resting in water (SRW) were measured. The targeted high intensity and dynamic recovery HR formula as follows: [(206.9 − (0.67 × age) − (HR_SRL−HR_SRW) − HR_SRW)] × 75/50% + HR_SRW [47]. The participants were led by a professional certified aquatic fitness instructor for exercise and were protected by two research assistants in the water. The HR, RPE and BP were recorded immediately after exercise SRW, 10 min after recovery SRL, and 45 min after recovery SRL. The participants went into the quiet room again to measure baPWV 60–120 min after the testing was completed. The depth of water was 98 cm, and the water temperature was 30–33 °C. The room temperature was 25–27 °C, and the air humidity was 65–75%. The experimental procedure was shown in Figure 1.

### 2.3. Exercise Intervention

The 36-min session includes warm-up (5 min), twelve trails of 30 s upright aquatic HIIJ (18 min, including counter movement jump, lunge jump, jumping opening legs and arms, single leg hop and twelve trails of dynamic recovery), three sets of resistance exercises (3 min, including latissimus dorsi exercise and triceps exercise), group dynamic recovery (5 min) and cooling down (5 min). The jumping intensity is 75% HRR, and the dynamic recovery intensity is 50% HRR. Latissimus dorsi resistance exercises and triceps resistance exercises were carried out as fast as possible for 20 s, each followed by a 10-s resting.

### 2.4. ACE Genotyping

The oral mucosal cells were collected to extract DNA. DNA fragments containing ACE gene polymorphism were sequenced using polymerase chain reaction (PCR). The steps of PCR are as follows: the DNA segments were denatured at 95 °C for 1 min, bonded at 67 °C for 40 s and extended at 72 °C for 2 min. This cycle was repeated 30 times [48]. After amplifying I/D polymorphic fragments, electrophoresis analysis was performed with 6% polyacrylamide gel. The segments were stained with ethidium bromide (40 min) and measured and photographed with ultraviolet light with a wavelength of 260–280 nm, as shown in Figure 2.

The ACE gene polymorphism testing produces three possible genotypes, including two homozygous (DD and II) genotypes and one heterozygous (ID) genotype. The primers used in the PCR allow amplification of sequences of the DD genotype with 190 base pairs (bp) and the II genotype with 490 bp. The primer pair sequences used are hECAf (5′-CTG GAG ACC ACT CCC ATC CTT TCT-3′) and hECAr (5′-GAT GTG GCC ATC ACA TTC GTC AGA T-3′). The type I allele of the ID genotype may be suppressed and is not easily amplified during PCR amplification, so the ID genotype may be misjudged as a DD genotype. All samples interpreted as DD genotype need to go through highly specific type I allele testing. The specific primers used are 5′-TGG GAC CAC AGC GCC CGC CAC TAC-3′ sense and 5′TCG CCA GCC CTC CCA TGC CCA TAA-3′ antisense.

### 2.5. Measures

#### 2.5.1. Body Composition

Body composition was measured using the Inbody 230, with participants fasting for at least 2 h and not exercising before the measurement. The participants wore light clothing and removed their shoes, socks, and metal objects for measurement with the hands and feet in full contact with the electrodes.

#### 2.5.2. Heart Rate

The HR was recorded with the Garmin heart rate monitor (Garmin 920 XT, Garmin Ltd., Schaffhausen, Switzerland) with the heart rate belt under the chest and the signal transmitter on the sternum.

#### 2.5.3. Blood Pressure

The BP was measured with the BP automatic device (HEM7210, Omron Co., Ltd., Kyoto, Japan). The measurement position was the left upper arm. The cuff was kept at the same horizontal height as the heart. The participants needed to dry the water stains on their arms in time while exercising in water. They were required to keep quiet during the measurement.

#### 2.5.4. Brachial-Ankle Pulse Wave Velocity

The baPWV was measured with the AS automatic device (Colin VP-1000, Colin Co., Ltd., Komaki, Japan). After lying down, the participant wore the cuff on the upper arm and ankle. The wrists and the fourth section of the sternum of the subject were disinfected with alcohol swabs. The ECG sensors were clipped on the left and right wrists. The heart sound sensor was placed on the fourth sternum and was compacted with the weight bag of the tester. Measurements started after the participants had been lying quietly for 15 min. The time interval between the wavefront of the brachial waveform and that of the ankle waveform was defined as the time interval between the brachium and ankle (∆Tba). The baPWV was calculated from ∆Tba, the path length from the suprasternal notch to the brachium (Lb), and the path length from the suprasternal notch to the ankle (La). To be more specific, baPWV = (La−Lb)/ΔTba.

### 2.6. Statistical Analysis

Data are expressed as mean and standard error (Mean ± SE). The SPSS version 23.0 software (SPSS Inc., Chicago, IL, USA) was used for statistical analysis. The chi-square test was used to test the differences in exercise habits, HTN, circulatory system and skeletal system disease prevalence, etc., between postmenopausal women carrying the DD genotype and those carrying the II/ID genotype. Analysis of covariance was used in examining the differences in the mean values of the BP and baPWV that are related to the effect of hypertension. One-way repeated measurement-ANOVA was used to determine the changes of baPWV of all participants after exercise. A two-factor mixed ANOVA design was used to determine the differences in changes of BP, HR, RPE, baPWV at each measurement time point based on the values at the resting state between postmenopausal women with the DD genotype and those with the II/ID genotype. Statistical significance was set a prior at α = 0.05 (*p* < 0.05). After acquiring test data from the 33 participants (3 DD genotype, 30 II/ID genotype), the BP and baPWV effect size were 0.52, 0.61, and 0.23, respectively. The analysis was performed using G Power software (Heinrich Heine University Düsseldorf, Düsseldorf, Germany) with a power set of 0.8. The minimum targeted sample sizes were 70. We recruited 101 postmenopausal women as participants. A total of 73 participants completed the study. We calculated the effect size from 73 participants. The effect size was from 0.45 to 0.69 (average 0.56) with BP and baPWV.

## 3. Results

### 3.1. Comparison of the Characteristics and Exercise Intensity for Different Genotypes

There were no significant differences between the DD genotype and the II/ID genotype in age, weight, BMI, body fat percentage, skeletal muscle mass, and prevalence of diseases (Table 1). The change of HR between stand resting on land and in the water of the DD genotype was significantly greater than that of the II/ID genotype (*p* < 0.05). There were no significant differences between the two groups of genotypes in HRR, HR, RPE and the movement numbers of aquatic jumping exercises and muscle resistance exercises (Table 2).

### 3.2. Comparison of BP for Different Genotypes during HIIJ Trial

The change in SBP immediately after exercise of the DD genotype was significantly greater than that of the II/ID genotype (*p* = 0.038). The change in SBP at 45 min after the recovery of the DD genotype was significantly smaller than that of the II/ID genotype (*p* = 0.026). There were no significant differences in the changes of DBP and MAP at each stage between the DD genotype and the II/ID genotype (Table 3).

### 3.3. Comparison of baPWV for Different Genotypes before and after Exercise and the Change of baPWV after the Exercise

The left and right baPWV of all participants increased significantly after exercise (*p* = 0.020, 0.011). The right baPWV of the II/ID genotype increased significantly after exercise (*p* = 0.047). The left and right baPWV and the change of baPWV after exercise were no significant differences between the DD genotype and the II/ID genotype (Table 4).

## 4. Discussion

The major finding of the present study was that the SBP increment of the DD genotype of postmenopausal women immediately after exercise was greater than that of the II/ID genotype. Moreover, the DD genotype of postmenopausal women had a smaller decline in SBP than the II/ID genotype at 45 min after recovery. The arterial stiffness in all postmenopausal women increased after exercise, and the increase in the DD genotype participants was about 2 to 3 times that of the II/ID genotype participants. The HR of the DD genotype participants decreased more significantly after entering the water from land.

### 4.1. Comparison of Post-Exercise BP Response in Postmenopausal Women with Different Genotypes

The present study demonstrated that immediately after HIIJ in the water, the SBP increment of the DD genotype participants was significantly greater than that of the II/ID genotype participants. These findings are consistent with Freire et al. (2015) who adopted 50%, 75%, and 100%10 RM strength exercises. Their results showed that the DD genotype had a significantly greater SBP incremental response immediately after each exercise than the II or ID genotype [37]. This result may be attributed to the higher concentration of ACE in the DD genotype and the release of ACE by exercise stimulation. Ang II stimulates the increase in BP. The Ang II concentration in the DD genotype was twice that of the II/ID genotype [33]. Moreover, exercise causes sympathetic stimulation and enhanced activity of the renin-angiotensin-aldosterone system [49], which increases the activity of ACE in the DD genotype, thus increasing the conversion of Ang I into Ang II [33]. Ang II increases renal sodium and fluid reabsorption by releasing aldosterone, and arterial BP rises [32,33]. Therefore, the BP increment immediately after exercise is greater in the DD genotype. This result may also be attributed to the higher concentration of Ang Ⅱ that blocks the decrease in vasodilation induced by bradykinin. Bradykinin causes vasodilation by releasing prostacyclin, nitric oxide, and endothelium-derived hyperpolarizing factors. The high concentration of Ang Ⅱ reduces the effect of bradykinin, which reduces vasodilation [50,51]. This underlying mechanism may also affect the PEH effect.

PEH is currently the main known mechanism for lowering BP after exercise. Exercise causes the improvement of vascular endothelial function mediated by NO, vasodilation, post-exercise heart rate, cardiac output decrease, and lowered pressure of arterial blood on the vascular wall, which causes PEH [27]. The present study also demonstrated that the SBP reduction in the DD genotype at 45 min after recovery was significantly smaller than that of the II/ID genotype (*p* = 0.048), i.e., the SBP decline of the II/ID genotype population was greater after aquatic HIIJ. Previous studies have observed that the II/ID genotype has higher NO release than the DD genotype after incremental exercise. Thus, the II/ID genotype has a better BP-lowering effect [52]. The higher concentration of Ang II in DD genotype individuals affects the bioavailability of vascular NO, resulting in affects the effect of vasodilation and BP lowering [53]. Therefore, compared with the DD genotype, the SBP decrease is more significant in the II/ID genotype after exercise. These results are consistent with Santana et al. (2011) [38] and Goessler et al. (2015) [34]. Santana et al. (2011) reported that the SBP decrease was more significant in the II/ID genotype than in the DD genotype 1 h after exercise (−7.4 ± 8.4 vs.−2.0 ± 3.6, *p* < 0.05) [33]. Previous studies indicated that individuals carrying the I allele (II/ID) are less likely to develop HTN and more likely to have PEH due to the lower ACE levels of the II/ID genotype compared to the DD genotype [33].

In addition, the present study also demonstrated that the DBP, MAP, and their changes were not significantly different between the genotypes immediately after exercise, immediately after recovery, 10, or 45 min after recovery. Results in the present study conform well to the findings by Santana et al. (2011) [38]. Santana et al. (2011) showed that the DBP, MAP, DBP and MAP changes of the DD genotype and II/ID genotype were not significantly different at 1 h after 20-min incremental to exhaustion exercise and 20-min 90% anaerobic threshold exercise [38]. Santana et al. (2011) have indicated that the ACE genotypes may affect mainly the SBP response after exercise but barely DBP and MAP [38]. DBP is the arterial pressure during ventricular diastole. Previous studies indicated that the increase in DBP after exercise was significantly smaller than that of SBP [53], i.e., the effect of exercise on DBP was relatively smaller, which may be the main reason for the insignificant DBP changes. In the calculation formula for MAP, MAP = DBP + 1/3 (SBP–DBP), the proportion of SBP is relatively small, and the proportion of DBP is relatively large. The insignificant change in DBP after exercise may be the reason for the insignificant change in MAP. Studies have also suggested that since MAP was responsible for maintaining the normal operation of the blood circulatory system and normal blood flow, there may be a protection mechanism to keep MAP relatively stable [38]. There are also inconsistencies between the results of the present study and other previous studies. Friedl et al. (1996) researched with an incremental to exhaustion aerobic power cycling test and found that the DBP of the DD genotype was significantly higher than that of the II/ID genotype at maximum exercise intensity [54]. The DBP of the DD genotype was still significantly higher at 3 min after exercise. The reason for the higher DBP in the DD genotype may be the way of strength exercise. Previous studies showed that strength exercise induces muscle contraction and edema, which reduces the diameter of blood vessels in the muscles. As a result, the peripheral resistance of blood circulation increases, which leads to an increase in arterial BP. This BP increasing response is especially greater with higher intensity strength exercise [55,56]. Previous studies have indicated that the DD genotype is associated with higher BP and HTN risk [34,37]. It is worth noting that there is no difference in SBP, DBP, and MAP between the ACE DD genotype and the II/ID genotype before exercise in the present study. The results were inconsistent with the previous studies [37]. This result may be attributed to the positive effects of a high percentage of regular exercise in the DD genotype (91.7%). The previous studies have demonstrated that regular exercise training reduces sympathetic nerve activity, increases plasma nitric oxide [10,20], improves vascular endothelial function [27], and enhances baroreflex sensitivity [57,58]. Therefore, the resting blood pressures were similar between the ACE DD genotype and II/ID genotype in the present study.

### 4.2. Comparison of Post-Exercise baPWV in Postmenopausal Women with Different Genotypes

As an indicator is often used to determine the progression, morbidity, and mortality risk of cardiovascular diseases [59,60,61]. Studies indicated that for every 100 cm/s increase in arterial PWV at the resting state, the risk of cardiovascular events increases by 12 to 14%, and the mortality risk from cardiovascular diseases increases by 13 to 15% [62]. The hemodynamic changes caused by exercise stimulation reflect the risk of vascular abnormalities and cardiovascular diseases [63]. The present study showed that the left and right side baPWV of all participants at 60 to 120 min after exercise was significantly higher than that before exercise (*p* = 0.020, 0.011). The results of the present study were consistent with the findings of many previous studies. Perissiou et al. (2018) [44], Müller et al. (2015) [40], and Yamato et al. (2016) [64] observed that aerobic/resistance exercise caused a significant increase in PWV. Specifically, Perissiou et al. (2018) indicated that PWV at 90 min after 40% peak power output (PPO) moderate-intensity aerobic exercise and 90 min after 70% PPO intermittent exercise was significantly higher than before exercise [44], which may be attributed to the sympathetic stimulation caused by exercise. Sympathetic stimulation increases the vascular tone against peripheral resistance to help transport arterial blood to peripheral blood vessels and tissues and ensure the body’s supply of oxygen and energy materials. The results of the present study were also inconsistent with findings from some studies. Yamato et al. (2016) found that PWV immediately after stretching exercises was significantly reduced and returned to the baseline level within 60 min [64]. The reason for that probability was the exercise modes. Studies showed that stretching exercises promote the enhancement of parasympathetic nerve activity and the decrease in sympathetic nerve activity, which promotes the decrease in vascular tone and vascular sclerosis. The age of the participants was the second reason for the lowered PWV in the study by Yamato et al. (2016) [64], the average age of the participants was 21 years old, while that in our study was 62.3 years. As in old-aged groups was related to the vascular structure caused by aging compared with young-aged groups [65]. Aging causes nitric oxide-mediated endothelial cell dysfunction [66], which leads to thick and hard arterial walls, and an increase in the number and size of smooth muscle cells [6]. The degree of hardening gradually increases with age, which reduces the arterial buffering function [67].

### 4.3. Comparison of Post-Exercise HR in Postmenopausal Women with Different Genotypes

Resting HR is an important indicator of cardiac function and an important parameter for exercise intensity. The present study indicated that after entering the water from the land, the resting HR of postmenopausal women with the DD genotype and the II/ID genotype was significantly lowered by 9.6 bpm and 7.2 bpm, respectively. The findings of the present study support previous research results. Previous studies have pointed out that HR should be lowered by 7 to 13 bpm than in land environments when exercising in water, and some studies have suggested lowering HR by 11 to 17 bpm [68,69]. Lowering HR is mainly due to the hydrostatic pressure when standing in water. The hydrostatic pressure increases the venous return and decreases the peripheral blood volume. As a result, the end-diastolic volume and stroke volume increase, thus decreasing HR [70,71,72]. The lowered HR may also be attributed to the neuromodulation induced by the water environment. The increase in blood reflux in the water environment causes a great burden on the heart. The baroreflex mechanism of the atria and arteries increases the vagus nerve activity and parasympathetic nerve activity and decreases the sympathetic nerve activity, which is also a possible reason for the decline in HR [73,74]. However, the present study indicated that the decline in resting HR in the two genotypes was smaller than that in previous studies, which may be due to water depth. In the previous studies, the water depth was mostly at the xiphoid, while that in our study was approximately up to the navel (98 cm). A greater water depth means a greater hydrostatic pressure and a greater drop in HR. Thus, the HR decrease in the present study was relatively smaller. It is worth noting that no research explored the differences in HR between different ACE genotypes in water environments until now. In addition to, the present study also found that the resting HR decrease in the DD genotype after entering the water from the land was greater than that of the II/ID genotype (−9.6 ± 4.14 vs. −7.2 ± 4.12 bpm, *p* = 0.046). The reason for this may be attributed to the baroreflex mechanism. It is known that the resting HR of the two genotypes decreased significantly after entering the water, while the hydrostatic pressure of the water increased blood pressure. The present study found that the resting SBP increased significantly (118.0 ± 15.93 vs. 123.5 ± 16.64 mmHg, *p* = 0.000) after entering the water from land. The elevated blood pressure activates the baroreflex response, which inhibits sympathetic nerve activity. The blood pressure response related to the baroreflex is particularly reflected in the DD genotype [57,58]. The higher blood pressure response of the DD genotype inhibits the sympathetic nerve activity, which may cause increased parasympathetic nerve activity and lower HR [43,58]. Therefore, the DD genotype showed a greater reduction in HR. Thus, in the design and practice of aquatic exercise courses, special consideration should be given to the HR response of the DD genotype and the potential cardiovascular workload to avoid cardiovascular risks.

### 4.4. Limitations

One of the limitations of the present study was that the baPWV detection was from 60 to 120 min after exercise. Since the participants need time to wash and change clothes and lie down and rest for 15 min before collecting the data, it is impossible to collect the baPWV within 60 min after exercise. Therefore, it is impossible to obtain the vascular stiffness changes within 60 min after exercise in postmenopausal women. In addition, the insufficient sample size of DD genotype participants was one of the limitations of the present study. The other limitation of the present study was that the circulating renin, AngⅡ, bradykinin, NO, and other related parameters were not measured. Thus, the related mechanisms of blood pressure, arteriosclerosis, and HR were inferred. We suggested should test them in the future to further explore and prove the mechanisms.

### 4.5. Implications

Studies have demonstrated the benefits of aquatic exercise on cardiovascular health in postmenopausal women [75,76]. However, several studies also showed that the blood pressure responses in aquatic exercise were higher than in dryland exercise [77,78]. Although the blood pressure responses in the present study were consistent with the previous dryland-exercise studies [34,38], our study was the first study to demonstrate that DD genotype participants had higher blood pressure response and an increasing trend of baPWV than II/ID genotype one after aquatic exercise. Postmenopausal women are at high risk of cardiovascular diseases. It is well known that the BP response after aquatic exercise is higher than on dryland [53]. Based on safety, we recommended that BP should be monitored when postmenopausal women engage in aquatic exercise [79]. Furthermore, our results showed that the DD genotype participants have higher BP responses than II/ID genotype after aquatic exercise. Therefore, BP should be more often monitored for postmenopausal women with the DD genotype.

## 5. Conclusions

The present findings demonstrated that the SBP increase immediately after exercise was significantly greater in postmenopausal women with the DD genotype than those with the II/ID genotype, SBP decreased at 45 min after exercise was significantly smaller. These findings highlight the potential risk of aquatic high-intensity intermittent jumping exercises on blood pressure elevated in postmenopausal women with the DD genotype. In addition, the II/ID genotype gains more benefits in lowering blood pressure after an aquatic HIIJ exercise program.

## Figures and Tables

**Figure 1 ijerph-19-08985-f001:**
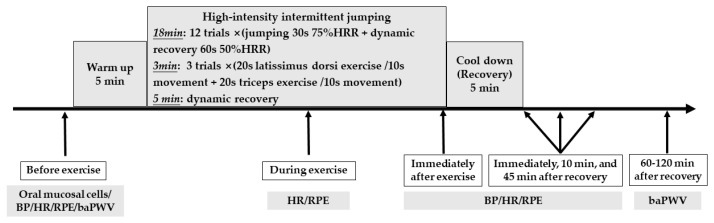
Experimental procedure. Note: HRR = heart rate reserve, s = seconds, BP = blood pressure, HR = heart rate, RPE = rating of perceive exertion, baPWV = brachial-ankle pulse wave velocity.

**Figure 2 ijerph-19-08985-f002:**
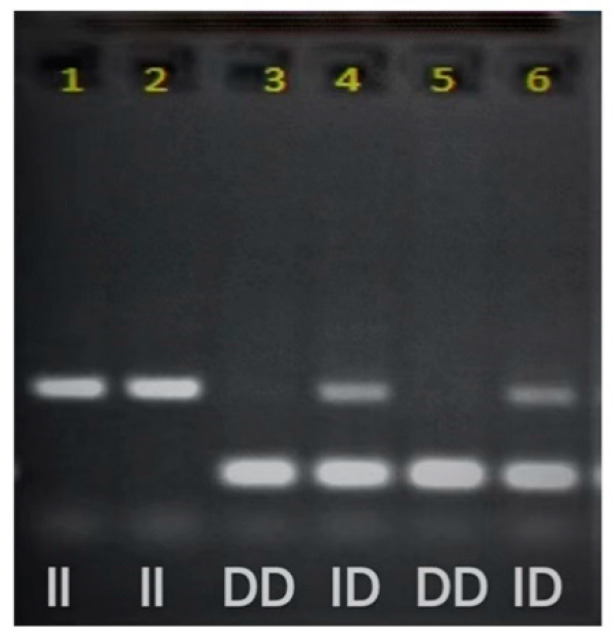
ACE polymorphism gene identification.

**Table 1 ijerph-19-08985-t001:** Characteristics of the participants (n = 73).

	DD Genotype (n = 12)	II/ID Genotype (n = 61)	*p*
Age	62.3 ± 1.45	62.2 ± 0.79	0.957
Weight (kg)	58.5 ± 2.23	57.2 ± 0.88	0.557
BMI (kg/m^2^)	23.5 ± 0.56	23.2 ± 0.34	0.728
Body fat (%)	32.6 ± 1.44	31.9 ± 0.71	0.663
Skeletal muscle mass (kg)	21.7 ± 0.77	20.8 ±0.29	0.692
Regular exercise percentage (%)	91.7	90.7	0.872
Proportion of people suffering from HTN (%)	16.7	19.7	0.809
Prevalence of circulatory system diseases (%)	16.7	9.8	0.489
Prevalence of bone and joints diseases (%)	25.0	13.1	0.293
Prevalence of endocrine diseases (%)	8.3	19.7	0.348
Prevalence of digestive system diseases (%)	16.7	14.8	0.866
Prevalence of ear and eye diseases (%)	25.0	23.0	0.878
Prevalence of respiratory system diseases (%)	8.3	1.6	0.194
Prevalence of cancer (%)	8.3	3.3	0.420

Note: Data presented are mean ± SE. Abbreviations: DD = deletion/deletion polymorphism, II/ID = insertion/insertion or insertion/deletion polymorphism, BMI = body mass index.

**Table 2 ijerph-19-08985-t002:** Comparison of HR, movement numbers and RPE with different genotypes.

	DD Genotype (n = 12)	II/ID Genotype (n = 61)	*p*
HR_SRL (bpm)	77.1 ± 1.84	73.9 ± 1.21	0.268
HR_SRW–SRL (bpm)	−9.6 ± 1.20	−7.1 ± 0.53	0.046
HR of aquatic jumping (bpm)	133.5 ± 1.69	134.8 ± 0.86	0.535
Intensity of aquatic jumping (%HRR)	75.2 ± 1.63	74.6 ± 0.64	0.732
RPE of aquatic jumping	4.1 ± 0.33	4.1 ± 0.14	0.915
HR of muscle resistance exercises (bpm)	134.3 ± 3.26	131.4 ± 1.66	0.481
Intensity of muscle resistance exercises (%HRR)	75.7 ± 4.12	70.9 ± 1.56	0.234
RPE of muscle resistance	4.4 ± 0.44	4.4 ± 0.18	0.967
Number of jumps (times)	421.8 ± 24.92	419.4 ± 6.97	0.899
Number of muscle resistance exercises (times)	157.5 ± 8.02	155.1 ± 2.29	0.712

Note: Data presented are mean ± SE. Abbreviations: DD = deletion/deletion polymorphism, III/ID = insertion/insertion or insertion/deletion polymorphism, SRL = stand resting on land, SRW = stand resting in water, HR = heart rate, HRR = heart rate reserve, RPE = rating of perceive exertion, bpm = bit per minute.

**Table 3 ijerph-19-08985-t003:** Comparison of BP for different genotypes during HIIJ trial.

		DD Genotype (n = 12)	II/ID Genotype (n = 61)	*p*
SBP (mmHg)	Before exercise	114.3 ± 3.52	118.7 ± 1.56	0.262
	Immediately after exercise	144.7 ± 4.55	138.7 ± 2.04	0.236
	Immediately after recovery	122.6 ± 3.88	121.2 ± 1.73	0.736
	10 min after recovery	113.4 ± 3.82	112.3 ± 1.69	0.795
	45 min after recovery	109.0 ± 3.36	102.5 ± 1.49	0.343
	Immediately after exercise-before exercise	30.8 ± 4.48	20.4 ± 2.00	0.038
	10 min after recovery-before exercise	−0.7 ± 3.28	−6.4 ± 1.45	0.117
	45 min after recovery-before exercise	−5.1 ± 3.26	−13.2 ± 1.44	0.026
DBP (mmHg)	Before exercise	70.0 ± 2.30	70.0 ± 1.02	0.977
	Immediately after exercise	67.4 ± 2.65	67.8 ± 1.18	0.901
	Immediately after recovery	66.6 ± 2.13	66.8 ± 0.95	0.914
	10 min after recovery	71.1 ± 2.50	70.8 ± 1.11	0.897
	45 min after recovery	68.3 ± 2.09	66.7 ± 0.93	0.501
	Immediately after exercise-before exercise	−2.6 ± 2.22	−2.5 ± 0.99	0.937
	10 min after recovery-before exercise	1.1 ± 1.50	0.8± 0.66	0.864
	45 min after recovery-before exercise	−1.8 ± 1.84	−3.2 ± 0.81	0.466
MAP (mmHg)	Before exercise	84.5 ± 3.20	86.3 ± 1.40	0.591
	Immediately after exercise	93.1 ± 2.68	91.2 ± 1.32	0.558
	Immediately after recovery	85.0 ± 2.72	85.0 ± 1.22	0.973
	10 min after recovery	85.0 ± 2.27	84.7 ± 1.39	0.921
	45 min after recovery	81.4 ± 2.77	79.7 ± 1.16	0.564
	Immediately after exercise-before exercise	8.6 ± 1.89	4.8 ± 1.27	0.203
	10 min after recovery-before exercise	0.4 ± 1.87	−1.6 ± 0.82	0.325
	45 min after recovery-before exercise	−3.2 ± 2.42	−6.6 ± 0.95	0.161

Note: Data presented are mean ± SE. Abbreviations: DD = deletion/deletion polymorphism, III/ID = insertion/insertion or insertion/deletion polymorphism, SBP = systolic blood pressure, DBP = diastolic blood pressure, MAP = mean arterial pressure.

**Table 4 ijerph-19-08985-t004:** Comparison of baPWV for different genotypes before and after exercise and the change of baPWV after the exercise.

		DD Genotype (n = 12)	II/ID Genotype (n = 61)	All Participants (n = 73)
baPWV (cm/s)	Before exercise_R	1477.7 ± 63.77	1436.1 ± 28.22	1442.1 ± 30.34
	After exercise_R	1533.3 ± 68.74 ^#^	1461.5 ± 30.42 ^*^	1472.0 ± 33.09 ^*^
	Before exercise_L	1482.0 ± 63.92	1438.6 ± 28.29	1444.8 ± 29.54
	After exercise_L	1553.0 ± 66.51	1459.7 ± 29.43 ^#^	1473.4 ± 32.36 ^*^
	Change_R	55.7 ± 28.20	25.4 ± 12.48	-
	Change_L	71.0 ± 28.24	21.1 ± 12.50	-

Note: Data presented are mean ± SE. Abbreviations: DD = deletion/deletion polymorphism, III/ID = insertion/insertion or insertion/deletion polymorphism, baPWV = brachial-ankle pulse wave velocity, R = right, L = left. * indicates that there is a significant difference before and after exercise (*p* < 0.05), # indicates the increasing trend after exercise (*p* < 0.1).

## Data Availability

The data used to support the findings of the present study are available from the corresponding author upon request.

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
