# Peer review of "Effects of Acute Aquatic High-Intensity Intermittent Exercise on Blood Pressure and Arterial Stiffness in Postmenopausal Women with Different ACE Genotypes"

_ijerph, 2022, doi:10.3390/ijerph19158985_

Round 1
Reviewer 1 Report
In this study, Zhou and colleagues compared exercise responses to aquatic high-intensity intermittent jumping (HIITJ) exercise in post-menopausal women with different angiotensin converting enzyme (ACE) genotypes. Subjects were screened into two groups- 1) those carrying the ACE deletion/deletion (DD) genotype and 2) those carrying either the insertion/insertion (II) or insertion/deletion (ID) genotype to test the hypothesis of that the DD genotype would receive more beneficial effects following exercise. Key findings include that the increase in systolic blood pressure immediately following exercise was significantly higher in the DD genotype and, that while brachial-ankle pulse wave velocity (baPWV) was increased in both groups following HIITJ, the increase was approximately 2-3 times greater in women with the DD genotype.
Specific comments
- This study is unique in that is uses an aquatics based exercise protocol. Beyond this, the findings from this study appear to be quite consistent with previous work in this area. Thus, the discussion should highlight how these findings advance the field and our knowledge in this area.
- Is this type of HIITJ exercise sustainable for women of this age and would consistent and repeated bouts of HIITJ over several months be predicted to the provide greater benefits for one group based on ACE genotypes?
- Why is baseline baPWV similar between different genotypes and what is the significance of the greater change following exercise in the DD genotype?
- Much of the data interpretation is via extrapolation and mechanistic insights appear to be limited by a lack of collection of blood samples for analysis of key parameters.
Author Response
Response to Reviewer 1 Comments
Point 1: This study is unique in that is uses an aquatics based exercise protocol. Beyond this, the findings from this study appear to be quite consistent with previous work in this area. Thus, the discussion should highlight how these findings advance the field and our knowledge in this area.
Response 1:
Thanks for your suggestion. We have added the relevant contents to the discussion content as follows: Studies have demonstrated the benefits of aquatic exercise on cardiovascular health in postmenopausal women (Colado et al., 2009; Ha et al., 2019). However, several studies also showed that the blood pressure responses in aquatic exercise were higher than on dryland exercise (American College of Sports Medicine, 2006; Dolbow et al., 2009). Although the blood pressure responses in the present study were consistent with the previous dryland-exercise studies (Santana et al. 2011; Goessler et al. 2015), our study was the first study to demonstrate that DD genotype participants had higher blood pressure response and an incresing trend of baPWV than II/ID genotype one after aquatic exercise. Postmenopausal women are at high risk of cardiovascular diseases. It is well known that the BP response after aquatic exercise is higher than that of on dryland (Asahina, et al., 2010). Based on the safety, it is recommended that BP should be monitored when postmenopausal women engaging in aquatic exercise (Chien, et al., 2015). Furthermore, our results showed that the DD genotype participants have higher BP responses than II/ID genotype after aquatic ex-ercise. Therefore, BP should be more often monitored for postmenopausal women with the DD genotype (Line 426-438).
Point 2: Is this type of HIITJ exercise sustainable for women of this age and would consistent and repeated bouts of HIITJ over several months be predicted to provide greater benefits for one group based on ACE genotypes?
Response 2:
Thank you for your question. The previous version of our manuscript indicated that postmenopausal women were restricted from participating in high-intensity intermittent exercise or aerobic exercise on dryland due to joint disease, insufficient muscle strength, fear of falling, and other factors (Lim et al., 2013). The aquatic HIITJ exercise can circumvent these restrictions, with buoyancy reducing 50-90% of the impact on joints, helping overcome the fear of falling and ensuring safety (Cole et al., 1994; Harrison., 1992) (Line 50-53). Therefore, the aquatic HIITJ exercise program is sustainable for middle-aged and older women.
Previous studies have demonstrated that the ACE II/ID genotype has a better blood pressure decrease than the DD genotype after a resistance exercise (Freire et al., 2015) (Line 277-280), 60-75%HRR aerobic exercise (Goessler et al., 2015) (Line 81-84), and 1 h after a maximal incremental ergometer test (Santana et al., 2011) (Line 305-307). Furthermore, ACE II and ID genotypes have also shown advantages in lowering blood pressure after long-term resistance exercise and aerobic exercise (Montrezol et al., 2019; Hagberg et al., 1999). Based on the results of these studies on dryland, we hypothesized that postmenopausal women with ACE II and ID genotypes had more significant benefits from water-based exercise training than those with ACE DD genotype. That needs further study to verify.
Extra Reference:
Montrezol, F.T.; Marinho, R.; da Mota, G.D.F.; D'almeida, V.; de Oliveira, E.M.; Gomes, R.J.; Medeiros, A. ACE gene plays a key role in reducing blood pressure in the hyperintensive elderly after resistance training. J. Strength Cond. Res. 2019, 33, 1119-1129.
Hagberg, J.M.; Ferrell, R.E.; Dengel, D.R.; Wilund, K.R. Exercise training-induced blood pressure and plasma lipid improvements in hypertensives may be genotype dependent. Hypertension. 1999, 34, 18-23.
Point 3: Why is baseline baPWV similar between different genotypes and what is the significance of the greater change following exercise in the DD genotype?
Response 3:
Thanks for your question. The baseline baPWV mean values were no statistically significant difference between DD genotype and II/ID genotype individuals in the present study. One of the reasons was probably due to participants' high percentage of regular exercise habits against the negative baPWV effect of DD genotype. Previous studies have shown that regular exercise training reduced sympathetic nerve activity, plasma ANG II, and central AT1 receptor expression and enhanced baroreflex sensitivity (Gao et al., 2007; Liu et al., 2000; Liu et al., 2002; Mousa et al., 2008). Therefore, regular exercise may further decrease the baPWV level to induce that baseline baPWV was similar between DD and II/ID genotype participants. Furthermore, The baseline SBP and DBP were also similar between the two groups. Our study's standard deviation (SD) range of baPWV was from 200-300 cm/s. The significant deviation of the baPWV value was probably the second reason. The previous study also presented a significant deviation of the baPWV value. For example, the SD values of baPWV in elderly male subjects with normotensive and subjects with prehypertension were 202 cm/s and 401 cm/s, respectively (Gurunathrao et al., 2015). The SD value of baPWV of all older women was 303.3 cm/s (Shin et al., 2015).
The present study showed that right baPWV of II/ID genotype increased significantly after exercise, whereas the increased trend in DD genotype participants. The small sample size of DD genotype participants was probably the reason to explain those results. We added the description of the insufficient sample size limitation in the discussion section (Line 419-421). Although the right baPWV of DD genotype participants only had an increasing trend after exercise, the change of baPWV (after exercise minus before exercise) was about 2 to 3 times that of II/ID genotype participants (R mean: 52.7 vs. 25.5 cm/s; L mean: 65.8 vs. 21.2 cm/s). That meant aquatic jumping exercise would cause greater arterial stiffness of the DD genotype. In addition, the present study also indicated that the SBP increment of the DD genotype immediately after exercise was greater than that of the II/ID genotype.
Extra Reference:
Gao, L.; Wang, W.; Liu, D.; Zucker, I.H. Exercise training normalizes sympathetic outflow by central antioxidant mechanisms in rabbits with pacing-induced chronic heart failure. Circulation 2007, 115, 3095–3102.
Gurunathrao, P.S.; Manjunatha, A.; Kanti, D.K. Evaluation of arterial stiffness in elderly with prehypertension. Indian J. Physiol. Pharmacol. 2015, 59, 16-22.
Liu, J.L.; Irvine, S.; Reid, I.A.; Patel, K.P.; Zucker, I.H. Chronic exercise reduces sympathetic nerve activity in rabbits with pacing-induced heart failure: A role for angiotensin II. Circulation 2000, 102, 1854–1862.
Liu, J.L.; Kulakofsky, J.; Zucker, I.H. Exercise training enhances baroreflex control of heart rate by a vagal mechanism in rabbits with heart failure. J. Appl. Physiol. (Bethesda, Md.: 1985). 2002, 92, 2403–2408.
Mousa, T.M.; Liu, D.; Cornish, K.G.; Zucker, I.H. Exercise training enhances baroreflex sensitivity by an angiotensin II-dependent mechanism in chronic heart failure. J. Appl. Physiol. (Bethesda, Md.: 1985), 2008, 104, 616–624.
Shin, J. H., Lee, Y., Kim, S. G., Choi, B. Y., Lee, H. S., & Bang, S. Y. (2015). The beneficial effects of Tai Chi exercise on endothelial function and arterial stiffness in elderly women with rheumatoid arthritis. Arthritis Res. Ther. 2015, 17, 380.
Point 4: Much of the data interpretation is via extrapolation and mechanistic insights appear to be limited by a lack of collection of blood samples for analysis of key parameters.
Response 4:
We sincerely thank you for your professional review work on our article. Research funding and experimental protocol limitation for not collecting blood samples. We also suggested that collecting blood samples analyze the circulating renin, Angâ…¡, bradykinin, NO, and other related parameters in further research (Line 421-424).

Reviewer 2 Report
I think that the results presented in the paper entitled „Effects of Acute Aquatic High-Intensity Intermittent Exercise 2 on Blood Pressure and Arterial Stiffness in Postmenopausal 3 Women with Different ACE Genotyp” are valuable and interesting.
It is important to be aware that selected genotypes could be associated with an elevation in systolic blood pressure, which is especially crucial in the patients with hypertension. On the other hand, controlled exercises is necessery as preventation of cardiovascular disease and incidents of CVD.
I do however, three suggestions:
- verse 73: „The previous studies have shown that there was different blood pressure decreased in the people with different ACE genes after exercise”, in my opinion it should be corrected into „The previous studies have shown that there was different blood pressure altered in the people with different ACE genes after exercise”
- verse 268: „….which increases the concentration of ACE in the DD genotype…..” in my opinion should be „…..which increases the activity of ACE in the DD genotype….”, because activity not concentration of ACE converts AngI into powerful AngII.
- Please especially in abstract and conclusion section introduce the clinical value of received findings.
Author Response
Response to Reviewer 2 Comments
Point 1: 1. verse 73: “The previous studies have shown that there was different blood pressure decreased in the people with different ACE genes after exercise”, in my opinion it should be corrected into “The previous studies have shown that there was different blood pressure altered in the people with different ACE genes after exercise”.
Response 1:
Thank you for your suggestion. We have corrected the prescription as follows, “The previous studies have shown that there was different blood pressure altered in the people with different ACE genes after exercise” (Line 76-78).
Point 2: 2. verse 268: “….which increases the concentration of ACE in the DD genotype…..” in my opinion should be “…..which increases the activity of ACE in the DD genotype….”, because activity not concentration of ACE converts AngI into powerful AngII.
Response 2:
We have revised the contents in the discussion (Line 284).
Point 3: 3. Please especially in abstract and conclusion section introduce the clinical value of received findings.
Response 3:
We have added the clinical value of received findings in the abstract and conclusion sections, respectively, as follows, “These findings suggest that the aquatic exercise program has better effects in decreasing blood pressure in postmenopausal women with the II/ID genotype. DD genotype should pay attention to the risk of increasing blood pressure after HIIJ aquatic exercise” (Line 31-34). “These findings highlight the potential risk of aquatic high-intensity intermittent jumping exercises on blood pressure elevated in postmenopausal women with the DD genotype. In addition, the II/ID genotype gains more benefits in lowing blood pressure after aquatic HIIJ exercise program “ (Line 443-446).
